# Folded gastrulation and T48 drive the evolution of coordinated mesoderm internalization in flies

**Silvia Urbansky[†], Paula González Avalos[†], Maike Wosch, Steffen Lemke***

Centre for Organismal Studies Heidelberg, Heidelberg University, Heidelberg, Germany

**Abstract** Gastrulation constitutes a fundamental yet diverse morphogenetic process of metazoan development. Modes of gastrulation range from stochastic translocation of individual cells to coordinated infolding of an epithelial sheet. How such morphogenetic differences are genetically encoded and whether they have provided specific developmental advantages is unclear. Here we identify two genes, *folded gastrulation* and *t48*, which in the evolution of fly gastrulation acted as a likely switch from an ingression of individual cells to the invagination of the blastoderm epithelium. Both genes are expressed and required for mesoderm invagination in the fruit fly *Drosophila melanogaster* but do not appear during mesoderm ingression of the midge *Chironomus riparius*. We demonstrate that early expression of either or both of these genes in *C.riparius* is sufficient to invoke mesoderm invagination similar to *D.melanogaster*. The possible genetic simplicity and a measurable increase in developmental robustness might explain repeated evolution of similar transitions in animal gastrulation.

**\*For correspondence:** steffen.
lemke@cos.uni-heidelberg.de

[†]These authors contributed
equally to this work

**Competing interests:** The
authors declare that no
competing interests exist.

**Reviewing editor:** Siegfried
Roth, University of Cologne,
Germany

## Introduction

The evolution of shape and form is generally associated with changes of cell and tissue behavior during the course of development (*Carroll et al., 2013*). Such changes in morphogenesis can originate from modifications of molecular patterning, by adjusting the interpretation and integration of molecular patterning within and between cells, or by altering the translation of cellular decisions into changes of cytoskeleton and cell behavior, either alone or in combination (*Davies, 2013*). Understanding principal transitions in the evolution of morphogenesis thus requires the precise description of differences in cell and tissue behavior, the identification of coincident changes in gene activity, and a functional validation to support the link between molecular and morphogenetic divergence.

The evolutionary gain or loss of gene activity has been shown to be a major source of morphological innovation, in particular between closely related species and in systems where patterns of gene expression correspond to a direct phenotypic output like insect wing and body pigmentation (*Arnoult et al., 2013*; *Gompel et al., 2005*; *Prud'homme et al., 2006*; *Wittkopp et al., 2002a*, *2002b*). Well-studied examples of morphogenetic evolution have correlated differences in tissue and cell behavior to modifications in gene expression (*Cleves et al., 2014*; *Indjeian et al., 2016*; *Shapiro et al., 2004*), occasionally supported by experimental evolution (*Abzhanov et al., 2006*, *2004*), but rarely have analyzed morphogenesis in reference to a genetically well-understood developmental context (*Rafiqi et al., 2012*). It is currently not known whether major, macro-evolutionary changes of morphogenesis between distantly related species are driven through accumulative genetic tinkering, the recruitment or modification of pre-existing developmental modules in a switch-like fashion, or by the acquisition of entirely new gene function; neither is it understood how

**eLife digest** In animals, gastrulation is a period of time in early development during which a sphere of cells is reorganized into an embryo with cells arranged into three distinct layers (called germ layers). The process has changed substantially during the course of evolution and thus provides a great experimental system to investigate the genetic basis for differences in animal form and shape.

As an example, true flies use at least two different mechanisms to make the middle germ layer (the mesoderm). In both cases, the mesoderm is made up of cells that move inwards from the boundary of the outer germ layer. In midges and some other flies, these cells migrate individually, while in others including fruit flies, the cells move together as a sheet. Fruit flies and midges shared their last common ancestor 250 million years ago and although the genes that make the mesoderm in fruit flies are well understood, little is known about how the mesoderm forms in midges.

Urbansky, González Avalos et al. investigated which genes were responsible for the evolutionary transition between the different types of cell migration seen in flies. The experiments identified two genes – called *folded gastrulation* and *t48* – that seem to operate as a simple switch between the two ways that mesoderm cells migrate. Both of these genes are active in fruit fly embryos and are required for the group migration of mesoderm cells. However, the genes do not appear to play a major role in the movement of individual mesoderm cells in midges.

Further experiments demonstrate that switching on these genes in midge embryos is sufficient to invoke group mesoderm cell migrations similar to those seen in fruit flies. These findings show that it is possible to identify genetic changes that underlie substantial differences in animal form and shape over hundred million of years. The ease by which Urbansky, González Avalos et al. were able to switch between the two types of mesoderm migration might explain why similar transitions in gastrulation have evolved repeatedly in animals. The next step is to test this hypothesis in other animals.

many or what kinds of modifications are required to instruct transitions between different modes of morphogenesis.

To address these questions, we have studied gastrulation in the fruit fly *Drosophila melanogaster* and the midge *Chironomus riparius*, two fly species that shared their last common ancestor about 250 million years ago (*Wiegmann et al., 2011*). Their evolutionary history is characterized by an evolutionary transition between two distinct modes of cell internalization during gastrulation, i.e. mesoderm invagination in *D. melanogaster* and mesoderm ingression in *C. riparius* (*Figure 1*). Similar transitions between distinct modes of cell internalization have been observed repeatedly and in either direction during the evolution of metazoan gastrulation (*Leptin, 2005*; *Nielsen, 2012*; *Solnica-Krezel and Sepich, 2012*). In insects, the extremely fast and coordinated invagination of mesoderm cells constitutes a derived morphogenetic process that evolved from stochastic migration of individual cells (*Johannsen and Butt, 1941*; *Roth, 2004*). Mesoderm invagination in flies has been studied extensively in *D.melanogaster* and thus provides an excellent reference system (*Leptin, 2005*); mesoderm ingression is less well understood but has been previously reported for *C. riparius* as well as other species representing the most basal branches of flies (*Goltsev et al., 2007*; *Ritter, 1890*).

Mesoderm invagination in *D.melanogaster* builds on molecular patterning through the transcription factors Twist (Twi) and Snail (Sna) along the ventral midline of the blastoderm embryo. These transcription factors repress mitosis (*Grosshans and Wieschaus, 2000*; *Seher and Leptin, 2000*) and instruct specific shape changes in the cells of the mesoderm anlage (*Costa et al., 1994*; *Kölsch et al., 2007*; *Leptin, 1991*; *Manning et al., 2013*; *Martin et al., 2009*; *Rauzi et al., 2015*). Their positional information is relayed to the cytoskeleton by G-protein coupled receptor (GPCR) signaling through Folded gastrulation (Fog) as ligand, Mist as receptor, and Concertina (Cta) as associated Gα-unit. Upon binding of Fog to Mist, Cta is released and activates RhoGEF2, which is enriched at the cell apex of mesoderm cells through the transmembrane anchor T48. In turn, RhoGEF2 dependent activation of Rho1 invokes changes in cell shape through constriction

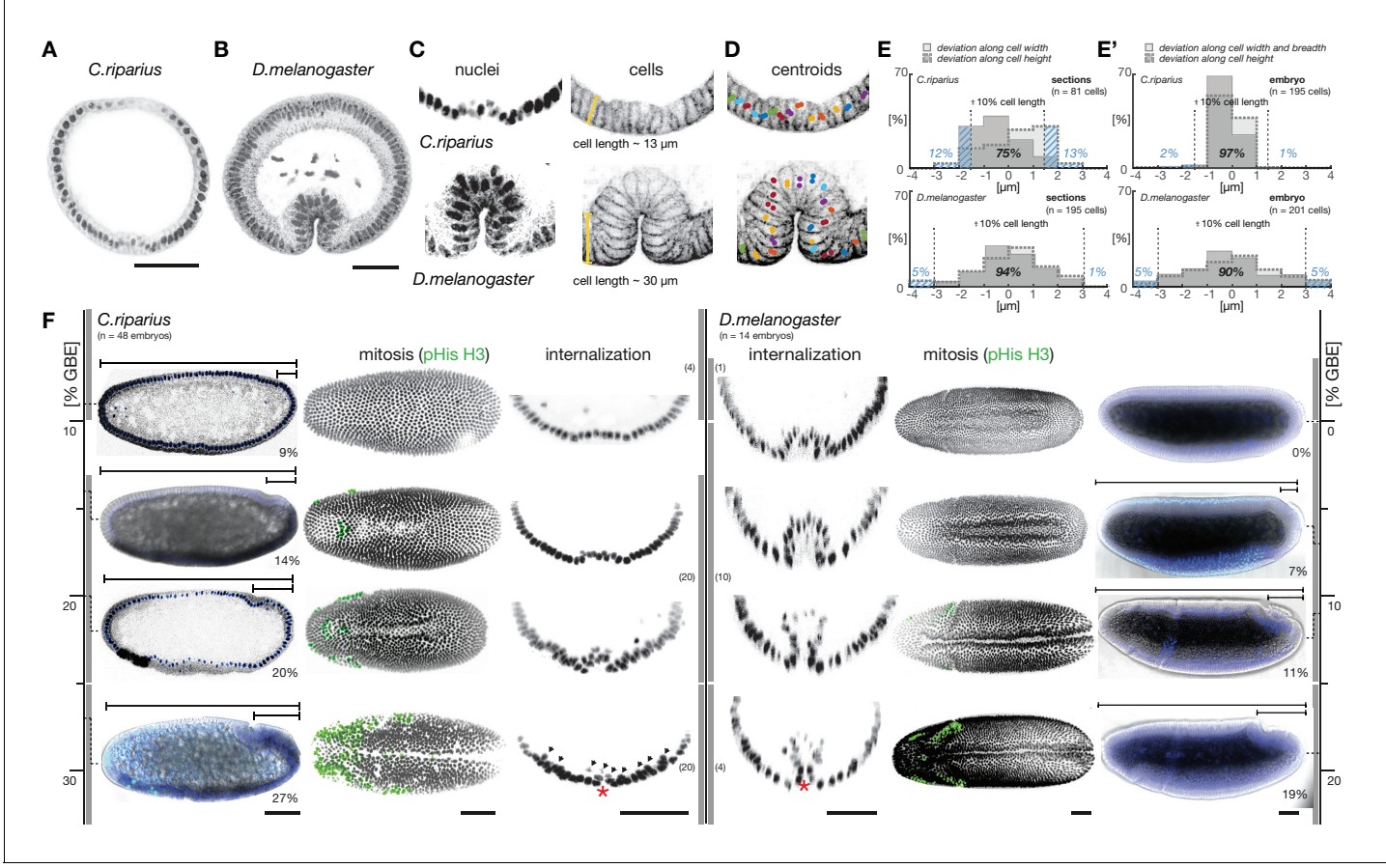

**Figure 1.** To assess gastrulation differences between *C.riparius* and *D.melanogaster*, nuclear position and embryonic axis elongation provide faithful proxies for cell position and stage of mesoderm internalization. (**A,B**) Mesoderm internalization shown for *C.riparius* (**A**) and *D.melanogaster* (**B**) in transversal embryonic sections stained for DNA (DAPI) and F-actin (phalloidin). (**C**) Cell positions as a readout of cell behavior were determined in transversal sections and whole embryos based on staining of DNA (cell nucleus) and F-actin (cell outline), cell length was measured along the yellow bar. (**D**) The centre of masses determined in embryo sections is indicated as circles overlaid on an F-actin stained micrograph (colors indicate identical cells). (**E,E'**) Differences in the approximation of the cell position by cell nucleus and outline were assessed as a deviation along cell height, width, and breadth for sections (**E**) and whole embryos (**E'**). The percentage of cells for which the two methods deviated by more than 10% cell length are indicated (blue dashed area). (**F**) Progression of germband extension (GBE) in *C.riparius* and *D.melanogaster* served as measure to stage mesoderm internalization. The extension was measured as displacement of the invaginating posterior midgut from the posterior pole in percent egg length (0% corresponds to the non-extended germband). Gastrulating embryos of *C.riparius* and *D.melanogaster* were classified according to the degree of GBE. To enable automated image segmentation routines to extract the ventral embryo surface, embryos in which the ventral opening was equal or smaller than one nucleus size (red asterisk) were considered as closed along the ventral midline. In *C.riparius*, embryos with less than 10% GBE showed no internalization, mesoderm internalization was observed in embryos showing 13 to 25% GBE, and embryos with more than 25% GBE were closed along the ventral midline. In these embryos, putative mesoderm nuclei were barely detectable (arrow heads), presumably because they were few and adhered closely to the ectoderm. In *D.melanogaster*, mesoderm internalization was observed in embryos before and after the onset of GBE, and embryos with more than 15% GBE were closed along the ventral midline (classes and number of embryos within class indicated as grey bars). For subsequent quantification of mesoderm internalization, *C.riparius* embryos were staged between 13–22% GBE, and *D.melanogaster* embryos between onset and 10% GBE. Mitosis in *C.riparius* was observed in spatially distinct and temporally successive domains similar as in *D.melanogaster* (**Foe, 1989**). Scale bars, 50 μm.

of the apical actomyosin network (reviewed in *Leptin, 2005*; *Manning and Rogers, 2014*). We used this defined morphogenetic module as reference to identify the genetic innovations that distinguish mesoderm invagination in *D.melanogaster* from ingression in *C.riparius*. We then asked whether the evolutionary transition between ingression and invagination could be experimentally reproduced, and finally tested whether the origin of mesoderm invagination in flies has been associated with putative developmental advantages.

# Results

An initial comparison of mesoderm internalization in *C.riparius* and *D.melanogaster* indicated differences in the mode of cell internalization, which were consistent with an evolutionary transition from ingression to invagination (*Figure 1A,B*). To establish whether differences between ingression and invagination could be studied by a careful analysis of cell positions in fixed tissue, we used staining of F-actin and DNA as a proxy for the cytoskeletal cell outline and nuclei, respectively (*Figure 1C,D*). In both species, we found that the position of mesoderm cells during progression of mesoderm formation could be approximated by their nuclei (*Figure 1E,E'*). To establish equivalent time points of mesoderm internalization during the courses of gastrulation in *C.riparius* and *D.melanogaster*, we compared the timing of mesoderm internalization in both species relative to a conserved and independent process of fly gastrulation, i.e., germband extension (*Anderson, 1966*). We found that relative to the onset of germband extension, mesoderm internalization started and ended earlier in *D. melanogaster* than in *C.riparius* (*Figure 1F*). In transversal sections of *C.riparius* embryos that were about halfway through internalization, mesodermal cells clustered on top of a shallow groove along the ventral midline, while at a comparable stage in *D.melanogaster* they formed part of a deep ventral furrow (*Figure 1A,B,F*). Taken together, the net behavior of mesoderm cells in the two species appeared to be either ingressive (*C.riparius*) or invaginative (*D.melanogaster*).

Notably, when we used cell position in transversal sections to assess the coherence of mesoderm cell behavior in individual embryos (see Materials and methods), we found mesoderm cell behavior to vary substantially along the anterior-to-posterior axis (*Figure 2A–B''*). These differences were more pronounced in *C.riparius* embryos than in *D.melanogaster* and indicated that the classification of cell internalization in fixed embryos as either ingressive or invaginative could not be based on qualitative measures of transversal sections alone. Thus, to account for variation in the mode of cell internalization within and between individuals, we established measures to quantitatively distinguish mesoderm ingression and invagination in whole embryos. Specifically, we quantified cell behavior during mesoderm internalization by using parameters that have been used to characterize this process in *D.melanogaster*. These included the total number of cells within the future mesoderm that underwent mitosis, the width and depth of the ventral furrow, the number of cells along the ventral midline that were internalized, a measurement of epithelial integrity, and the maximal depth at which an internalized cell could be observed (*Costa et al., 1993*; *Kam et al., 1991*; *McMahon et al., 2008*). Mitosis was detected by antibody staining against phosphorylated histone H3. All remaining features were quantified by approximating cell position based on the position of the nuclei relative to the computationally reconstructed egg and epithelial surfaces (*Figure 2C,D*, *Figure 2—figure supplement 1*). In the assessment of maximal cell internalization and furrow depth, the analysis based on nuclear cell position proved particularly beneficial as it was independent of cell orientation and avoided discrimination against distinct modes of mesoderm internalization. All features were evaluated for nine transversal sections of a ventral window for each embryo (see Materials and methods, *Figure 2—figure supplement 1*). While mitosis was rarely detected in either species during mesoderm internalization, wild-type embryos of *C.riparius* and *D.melanogaster* exhibited significant divergence in all other parameters (*Figure 2E*).

To address the extent to which each individual parameter contributed to the global differences in *C.riparius* and *D.melanogaster* mesoderm formation and detect biases in the analysis, we performed a principal component analysis of all the sections and parameters. We found three principal parameters that, regardless of intra-species variation in space and time, permitted the best discrimination between ingression and invagination: cell internalization, the depth of the ventral furrow and epithelial integrity (*Figure 2F–F'''*). Accordingly, invagination in *D.melanogaster* was characterized by high tissue integrity, a deep ventral furrow, and a high number of internalized cells, while ingression in *C. riparius* was characterized by low tissue integrity, a shallow ventral furrow, and a lower number of internalized cells. The number of internalized cells as well as the depth of the ventral furrow may be in part depended on the size of the mesoderm anlage and thus contain a species-specific component, whereas integrity was independent of the size of the mesoderm anlage. Collectively, the identified features made it possible to distinguish between ingressive and invaginative mesoderm formation in *C.riparius* and *D.melanogaster* precisely and quantitatively, despite variations within and between individuals.

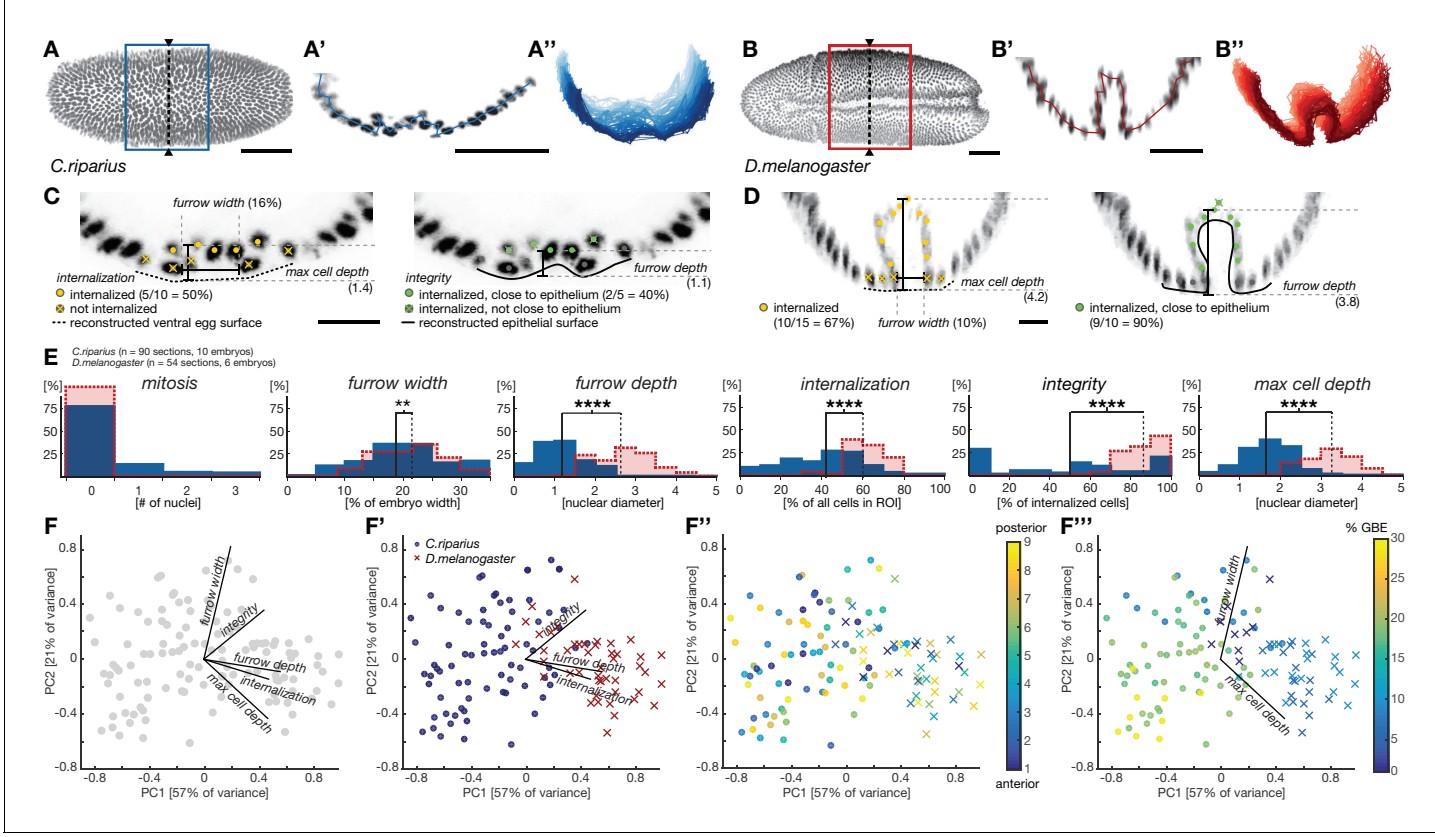

**Figure 2.** Quantitative analysis of mesoderm internalization in *C.riparius* and *D.melanogaster* characterizes differences between mesoderm ingression and invagination. (A–B'') Ventral view (A,B) transversal sections (A',B') and stacks of transverse centerlines (A'',B'') at comparable positions and corresponding stages of mesoderm internalization in *C.riparius* (A–A', blue; 17% GBE) and *D.melanogaster* (B–B'', red; 17% GBE). (A,B) Boxes indicate the region for which stack centerlines were calculated, arrowheads and dashed lines indicate the position of selected transversal section (A',B'). The stacks of transverse centerlines provide a visual measure for variation of cell behavior along the anterior-to-posterior axis of the embryo. (C,D) Exemplary quantitative analysis of cell behavior based on nuclear position as outlined in *Figure 2—figure supplement 1* on transversal sections of *C. riparius* (C) and *D.melanogaster* (D). Cell position based on segmented nuclei within a defined region of interest (ROI, see Materials and methods) are indicated as circles, lines are used to indicate the computationally reconstructed ventral egg surface and the outline of ventral epithelium. Automated classification of cells is indicated as color code (legend in C). Measures of furrow depth, and max cell depth were based on cell position and measured in nuclear diameter, furrow width was measured as a proportion of total embryo width (see Materials and methods). (E) Collective quantitative analysis of cell behavior for mesoderm internalization in transversal sections of *C.riparius* (blue) and *D.melanogaster* (red) embryos that separate mesoderm ingression from invagination. Medians of the distributions are indicated (solid line, *C.riparius*; dashed line, *D.melanogaster*). The significance of differences in the median was assessed by a Wilcoxon rank sum test. (F–F''') Biplots of a principal component analysis for all transversal sections of all analyzed *C.riparius* and *D.melanogaster* embryos. Shown is the contribution for each feature to the two first principal components (F), the separation of the two species and how they are most sensitively separated along integrity, furrow depth, and internalization (F'), the position of individual transversal sections along the anterior-posterior axis of the embryo (F''), and progression of gastrulation measured by % GBE (F''', *Figure 1F*). Furrow width and maximal cell depth align with the progression of gastrulation and were thus not considered as time-independent parameters of mesoderm internalization in *C.riparius* and *D.melanogaster*, respectively. Scale bars, 50 µm (A,A',B,B') and 20 µm (C,D). **p≤0.01; ****p≤0.0001.

The following figure supplement is available for figure 2:

**Figure supplement 1.** Cartoon summarizing image processing and analysis pipeline for the quantification of ventral cell behavior.

To identify the genetic basis for these differences, we used the established genetic network of *D. melanogaster* mesoderm internalization as a reference (*Figure 3A*) and asked how the expression and function of orthologous genes differed in *C.riparius*. In *D.melanogaster*, mesoderm internalization is orchestrated by two zygotic transcription factors, Twist (Twi) and Snail (Sna) (*Leptin and Grunewald, 1990*). Both genes are expressed along the ventral midline of the embryo (*Figure 3B–C'*), where they instruct zygotic expression of *fog* (*Figure 3D,D'*), *t48* (*Figure 3E,E'*), and *mist*

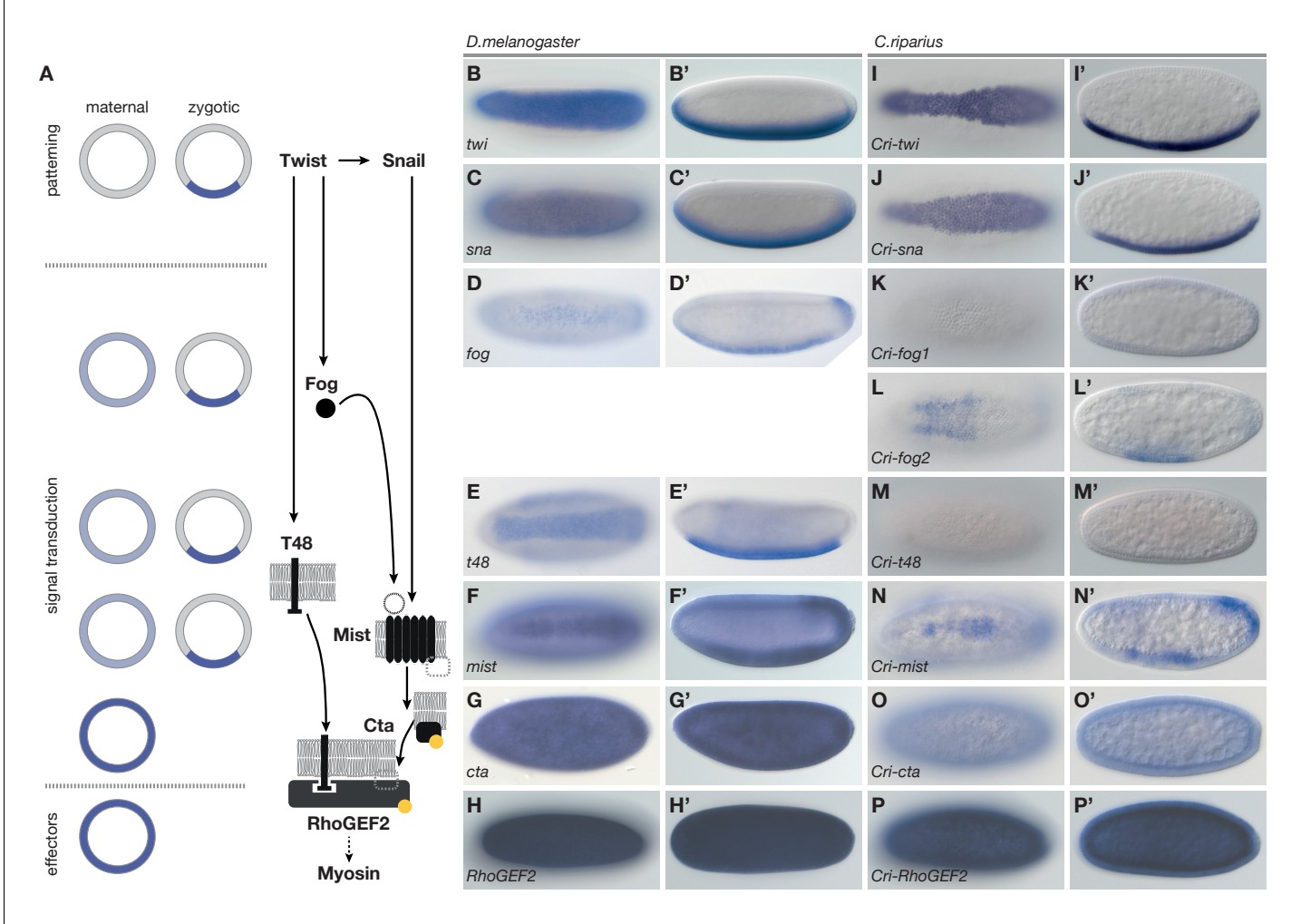

**Figure 3.** Mesoderm internalization in *C.riparius* lacks input by *fog* and *t48*. (**A**) Genetic regulation of mesoderm formation in *D.melanogaster* relays spatial instructions provided by the transcription factors Twi and Sna to RhoGEF2-dependent activation of non-muscle myosin via a GPCR signaling cascade with the ligand Fog, the G protein coupled receptor Mist, and its associated Gα subunit Cta; T48 functions as an apical anchor of RhoGEF2. The expression domains for each gene are indicated in schematized transversal sections for maternal (left) and zygotic (right) contribution (blue). (**B–P'**) The comparison of blastoderm expression patterns of *twi* (B,B'), *sna* (C,C'), *fog* (D,D'), *t48* (E,E'), *mist* (F,F'), *cta* (G,G'), and *RhoGEF2* (H,H') in *D. melanogaster* with the expression of the *C.riparius* orthologues Cri-twi (I,I'), Cri-sna (J,J'), Cri-fog1 (K,K'), Cri-fog2 (L,L'), Cri-t48 (M,M'), Cri-mist (N,N'), Cri-cta (O,O'), and Cri-RhoGEF2 (P,P') indicated potentially crucial differences in the expression of *fog* and *t48* in the future mesoderm.

The following figure supplements are available for figure 3:

**Figure supplement 1.** Phylogenetic occurrence of morphogenetic key players of *D.melanogaster* mesoderm invagination.

**Figure supplement 2.** Evidence for maternal contribution of selected GPCR signaling components in *D.melanogaster*.

(*Figure 3F,F'*), while *cta* and *RhoGEF2* are expressed ubiquitously (*Figure 3G–H'*) (*Costa et al., 1994*; *Kölsch et al., 2007*; *Manning et al., 2013*). Orthologues of *twi* and *sna*, *t48*, and *RhoGEF2*, along with genes encoding the GPCR ligand Fog, the receptor Mist, and the Gα subunit Cta, could be identified in *C.riparius* and in major families of winged insects where genome sequence data is available (Pterygota, *Figure 3—figure supplement 1*). The universal conservation of this gene set demonstrates its evolutionary importance and suggests that it originated over 400 million years ago (*Misof et al., 2014*).

To assess how members of this conserved set of genes contributed to gastrulation in *C.riparius*, we analyzed their expression (*Figure 3I–P'*) and function (*Figure 4*). We found that *Cri-twi* and *Cri-sna* were both expressed along the ventral midline of the blastoderm embryo (*Figure 3I–J'*). The knockdown of *Cri-twi* by RNAi resulted in embryos that lacked a visible ventral groove. Despite the knockdown of *Cri-twi*, nuclei of internalized cells could still be observed within the ventral-most 10–15% of the embryonic circumference (*Figure 4A,B*), which corresponded to a slightly narrower domain than that of wildtype *Cri-twist* expression. The knockdown of *Cri-sna* produced an abnormally thin ventral epithelium with spherical nuclei and a failure of proper mesoderm internalization (*Figure 4C*). Similar expression and phenotypes have been reported in experimental manipulations of *twi* and *sna* in *D.melanogaster* (*Leptin and Grunewald, 1990*), indicating that mesoderm cells receive and require essentially the same zygotic patterning information in *C.riparius* and *D.melanogaster*. Expression of *Cri-mist* along the ventral midline and in the domain of *Cri-twi* expression (*Figure 3N,N'*), as well as ubiquitous expression of *Cri-cta* (*Figure 3O,O'*) and *Cri-RhoGEF2* (*Figure 3P,P'*) was consistent with a RhoGEF2-mediated machinery of cytoskeletal regulation like in *D.melanogaster*.

Analysis of *fog* and *t48* gene expression patterns yielded significant differences between the two species (compare *Figure 3D–E'* with *Figure 3K–M'*). In *D.melanogaster*, both molecules are initially subject to maternal loading (*Figure 3—figure supplement 2*) (*Costa et al., 1994*; *Strutt and White, 1994*; *Zusman and Wieschaus, 1985*) and are later expressed zygotically along the ventral midline of blastoderm embryos in a Twist-dependent manner (*Costa et al., 1994*; *Kölsch et al., 2007*). In *C.*

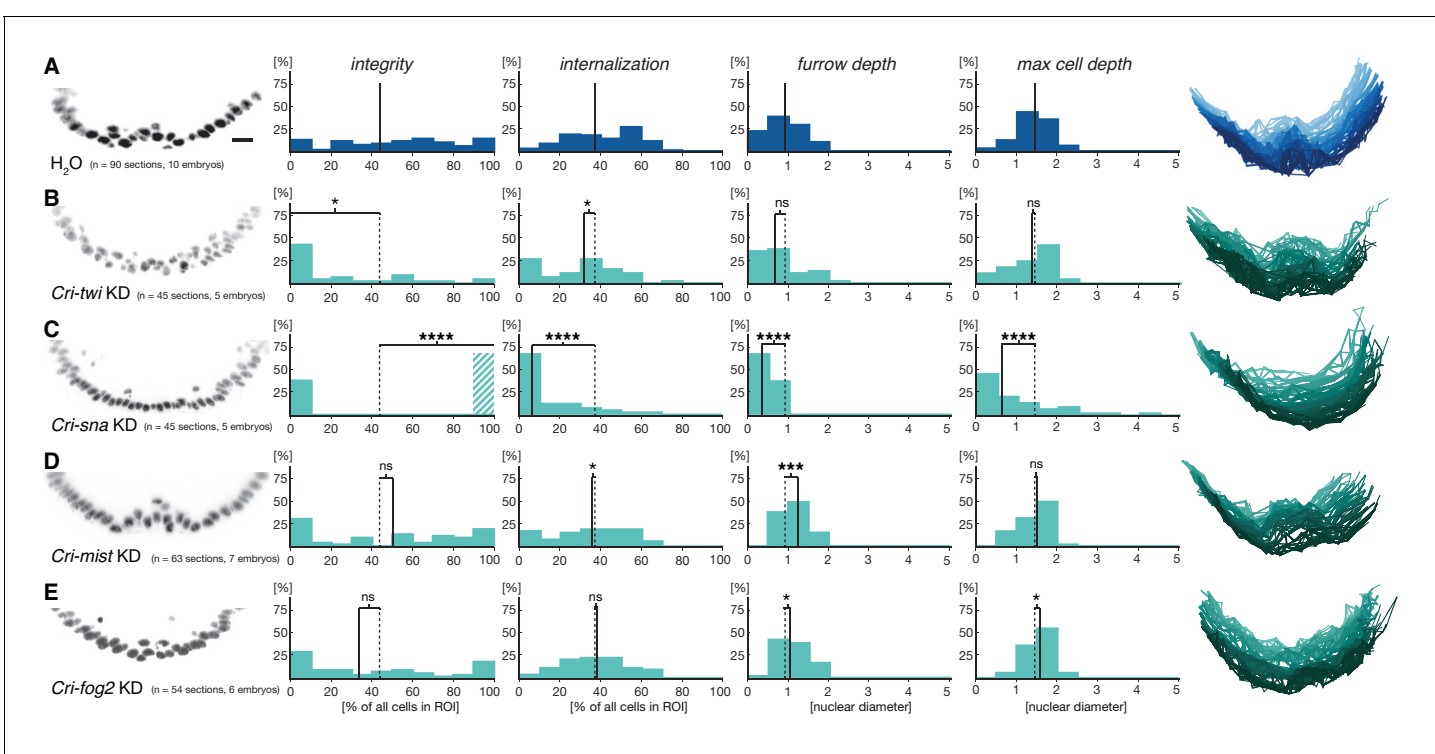

**Figure 4.** Mesoderm internalization in *C.riparius* depends on conserved patterning by *Cri-twi* and *Cri-sna*. (A–C) Compared to control injections with water (A), proper mesoderm internalization in *C.riparius* relied on patterning by *Cri-twi* and *Cri-sna*. Knockdown (KD) of *Cri-twi* showed a slight decrease in cell internalization but was mostly characterized by an absence of the shallow groove and seemingly random ingression throughout the ventral 10% of the embryonic circumference (B). Knockdown of *Cri-sna* produced embryos that failed to internalize the mesoderm, reflected by a highly significant decrease in furrow depth and max cell depth, as well as a highly significant increase in integrity when using a modified algorithm to account for lack of internalization (indicated as dashed bar, see Materials and methods) (C). (D–E) Knockdown of *Cri-mist* (knockdown efficiency 90%) appeared very similar to control injection, with a slight decrease in overall internalization and a small but significant increase in furrow depth (D); knockdown of *Cri-fog2* (knockdown efficiency 60%) was similar to control injections with a small but significant increase in furrow depth and maximal cell depth (E). Representative transversal sections, quantification of parameters, statistical analyses, and nuclear centerlines as in *Figure 2*. Scale bar, 20 μm. ns, p>0.05; *p≤0.05; **p≤0.01; ***p≤0.001; ****p≤0.0001.

*riparius*, *Cri-fog1* and *Cri-t48* transcripts were not detected until mesoderm internalization had been completed (*Figure 3K,K',M,M'*, *Figure 3—figure supplement 2*). A second fog orthologue, *Cri-fog2*, was expressed prior to mesoderm internalization in the central third of the embryo, in a domain broader than that seen for the expression of *Cri-twi* (*Figure 3L,L'*). This suggested that the GPCR ligand it encodes and, by extension, its receptor *Cri*-Mist, may not play the same prominent role in *C.riparius* mesoderm formation as they do in *D.melanogaster*. This interpretation was consistent with a visual inspection of embryos after RNAi knockdown of either gene. Following the quantitative analysis, we found that the median cell internalization and furrow depth differed in *Cri-mist* RNAi embryos from water injected embryos (*Figure 4D*). In *Cri-fog2* RNAi embryos, the difference was smaller and significant only for an increase in the furrow and maximal cell depth (*Figure 4E*). All effects after *Cri-mist* and *Cri-fog2* knockdown were substantially smaller than in *Cri-twi* and *Cri-sna* RNAi embryos and close to the sensitivity of our method (<10% of nuclear size), and we cannot exclude that either *Cri-fog* or *Cri-mist* do not contribute to mesoderm formation.

Based on the observed differences in *fog* and *t48* expression in *D.melanogaster* and *C.riparius*, we wondered whether the lack of substantial pre-gastrulation activity of both genes in *C.riparius* could be functionally linked to differences in the modes of mesoderm internalization between the two species. Specifically, we asked whether mesoderm invagination might be provoked in *C.riparius* by emulating Drosophila-like gene expression of *fog* and *t48*. We tested this hypothesis by injecting *in vitro* transcribed mRNA of the *D.melanogaster* genes *fog* and *t48* into *C.riparius* embryos at the early blastoderm stage. In response to this unspecific and early ubiquitous expression of *fog* and *t48*, *C.riparius* embryos exhibited an invagination of the mesoderm. This invagination was seen in several ways. First, we observed the formation of a distinct continuous ventral furrow, qualitatively comparable to that of *D.melanogaster* (*Figure 5A,B*). Then in particular, the induced invagination in *C.riparius* embryos was characterized by increased epithelial integrity and an increased furrow depth (*Figure 5B*).

To test whether the induction of mesoderm invagination in *C.riparius* required the activity of the *D.melanogaster*-specific protein sequence of *fog* and *t48*, we repeated the experiments with *Cri-fog* and *Cri-t48*. Following early ubiquitous expression of *Cri-fog* and *Cri-t48*, *C.riparius* embryos responded with an invagination of the mesoderm which resembled that seen with the *D.melanogaster* forms of *fog* and *t48* (*Figure 5C*), suggesting that a gain of early ubiquitous expression of one or both of these molecules was sufficient for the induction of mesoderm invagination. To test whether both *fog* and *t48* had been necessary for the evolution of mesoderm invagination, we tested whether a single factor alone was sufficient to induce invagination in *C.riparius*. Upon early ubiquitous expression of either *Cri-fog* or *Cri-t48* alone, *C.riparius* embryos displayed induced furrows overall, with increased epithelial integrity and deeper internalization very similar to the features observed in *Cri-fog* and *Cri-t48* double-injected embryos (*Figure 5D*). Differences were noted in the behavior of cells within individual embryos: these appeared less variable in embryos expressing *Cri-fog* than in embryos expressing *Cri-t48* (*Figure 5—figure supplement 1*). Our results indicate that *fog* and *t48* operate differently in transmitting instructions to the cytoskeleton, while at the same time demonstrating that gaining early expression of either gene alone was sufficient to invoke a change in the mode of mesoderm internalization. This interpretation was further supported by converse experiments in *D.melanogaster*. While the loss of either *fog* or *t48* activity in the mesoderm alone affects ventral furrow formation, it does not block it (*Costa et al., 1994*; *Kölsch et al., 2007*), but the combined knockdown of *fog* and *t48* functions in the ventral blastoderm abolished ventral furrow formation and resulted in ingression-like mesoderm cell behavior reminiscent of *C.riparius* wildtype development (*Figure 5E*).

In *D.melanogaster*, the expression of *fog* and *t48* both contribute to coordinated mesoderm invagination (*Figure 3A*) (*Manning and Rogers, 2014*). Our observations in *C.riparius* suggested that expression of either *Cri-fog* or *Cri-t48* was already sufficient for a morphogenetic response that appeared indistinguishable from the expression of both genes together. This seemingly redundant control of mesoderm invagination in *D.melanogaster* prompted us to speculate that features beyond cell internalization were associated with a tissue-wide invagination. In comparison with individual cells, a coherent epithelium provides increased mechanical stability (*Lecuit and Lenne, 2007*). In the context of mesoderm internalization, this stability of a coordinated invagination could provide robustness against physical perturbations that are widely present during fly gastrulation as the germband expands and the midgut invaginates (*Costa et al., 1993*). To test whether variation within *C.*

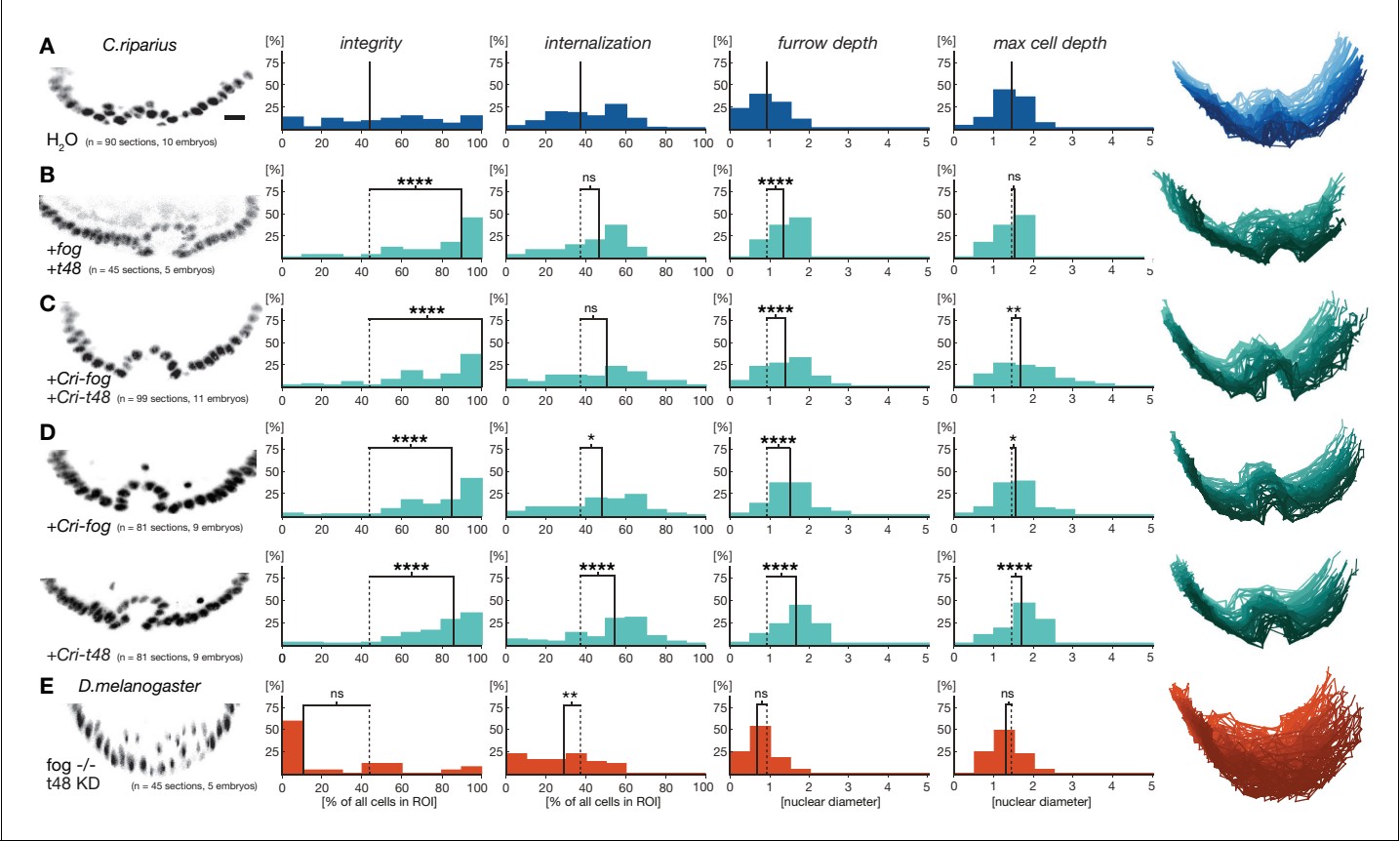

**Figure 5.** Ubiquitous expression of *fog* and *t48* promotes Drosophila-like mesoderm internalization in *C.riparius* embryos. (A) Mesoderm internalzation was characterized by a shallow furrow and little overall epithelial integrity in individual transversal section in control injected embryos (raw data as in *Figure 4A*). (B–D) Ubiquitous expression of *fog* and *t48* (B), *Cri-fog* and *Cri-t48* (C), or either gene alone (D) increased epithelial integrity. (E) In *D. melanogaster* embryos, knockdown of *t48* activity in embryos without ventral fog expression led to loss of epithelial integrity and tissue coherence comparable to *C.riparius* wildtype. Representative transversal sections, quantification of parameters, statistical analyses, and nuclear centerlines as in *Figure 2*. Scale bar, 20 µm. ns, p>0.05; *p≤0.05; **p≤0.01; ***p≤0.001; ****p≤0.0001.

The following figure supplement is available for figure 5:

**Figure supplement 1.** Comparison of individual embryos after either *Cri-fog* or *Cri-t48* injection suggests differences in the within-embryo variation of epithelial integrity.

*riparius* gastrulation could be reduced by early ectopic expression of *Cri-fog* and *Cri-t48*, we collected and compared all time-independent measures of mesoderm internalization between control-injected and *fog/t48* injected embryos (*Figure 6A*). Following the induced expression of *Cri-fog* and *Cri-t48*, we observed an overall decrease in the variability of phenotypes (*Figure 6B*). The decrease in the variability of the measured features was observed within embryos (*Figure 6C*) and may reflect the increase in cell coordination due to the induced change in the mode of gastrulation. Consistent with conceptual consideration of robustness (*Félix and Barkoulas, 2015*), we detected a decrease in variability also between embryos (*Figure 6C*), which indicated that *C.riparius* embryonic development became more robust against cellular mechanical perturbations.

To test whether the expression of *Cri-fog* and *Cri-t48* affected developmental timing, we counted how many cells had been internalized immediately after the onset of germband extension. Following the expression of *Cri-fog* and *Cri-t48*, we found a significant increase in the number of cells that were internalized (*Figure 6D*). An increased robustness against mechanical perturbation and developmental efficiency did not appear to trigger conflicts with subsequent embryonic muscle formation and enervation: after 48 hr, twitching and moving was observed in embryos injected with *Cri-fog*

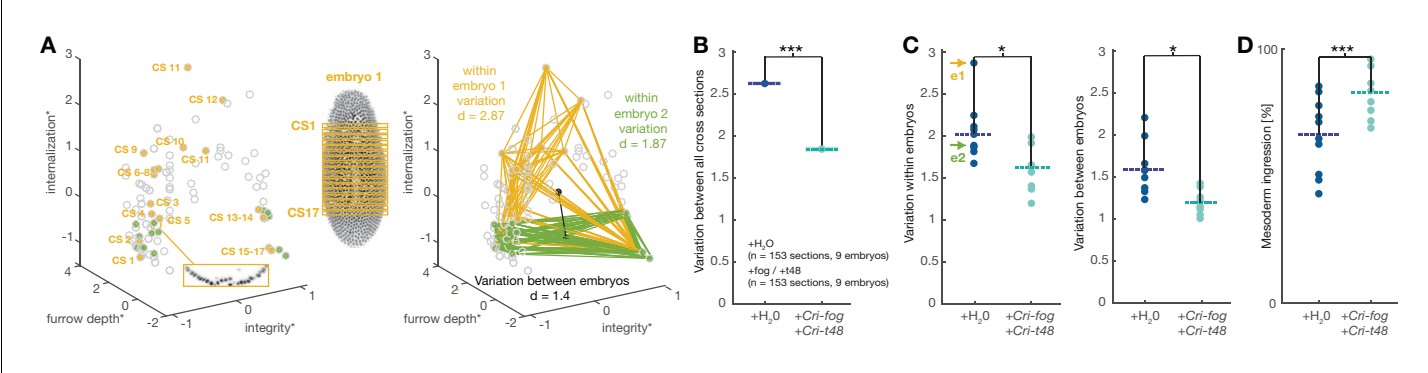

**Figure 6.** Experimentally induced invagination increases developmental robustness and efficiency of mesoderm internalization in *C.riparius*. Variation in gastrulation was assessed for each embryo by sampling the complete ventral ROI in seventeen transversal sections along the anterior-posterior axis. Each section was characterized by the time-independent parameters *internalization, furrow depth, maximal cell depth,* and *integrity* and takes one point in that 4D space. The similarity of cell behavior in different sections was calculated as the Euclidean distance between points. (**A**) Example of variation for two embryos in a reduced 3D data space (*internalization, furrow depth, integrity*; asterisk indicates normalization of parameter values by z-score normalization): each embryo was characterized by 17 cross sections (yellow and green points in 3D space), the Euclidean distance provided a measure for variation within the two embryos. Variation between embryos (black line) was computed as the Euclidean distance between the embryo centroids (black dots: calculated as the mean value for all sections in an embryo). (**B**) Variation across all transversal sections was significantly decreased after ectopic expression of *Cri-fog* and *Cri-t48*. (**C**) Both variations within individual embryos (values for embryos shown in A are indicated by arrows) as well as variation between embryos was decreased. (**D**) Ectopic expression of *Cri-fog* and *Cri-t48* increased the efficiency of mesoderm internalization and was measured as the percentage of internalized cells in the ROI at a comparable stage of GBE.

and *Cri-t48* at a frequency (40%; n=500) similar to control-injected embryos (44%; n=531). Our results thus indicate that, beyond a simple coordination of cell behavior, mesoderm invagination induced by *fog* and *t48* is associated with a measurable increase in the robustness and efficiency of gastrulation. Given the simplicity of the system and the extent of its fixation over long periods of evolutionary time, such traits could have provided an adaptive advantage in specific environments and contexts encountered by the predecessors of *D.melanogaster* after divergence from the last common ancestor shared with *C.riparius*.

## Discussion

Our analysis of *C.riparius* mesoderm morphogenesis has allowed us to associate two distinct modes of mesoderm internalization with specific differences in developmental gene expression. We could demonstrate that mesoderm ingression in *C.riparius* correlates at the genetic level with the absence of significant *fog* and *t48* activity in the early embryo. In comparison with mesoderm invagination in *D.melanogaster*, these results suggested that an evolutionary gain of early *fog* and *t48* activity may constitute a genetic switch to change the mode of fly gastrulation. We functionally tested this hypothesis by early ubiquitous expression of *fog* and *t48* in *C.riparius* embryos, and we found that the activities of either gene alone or both genes together were sufficient to invoke a Drosophila-like invagination of the mesoderm in *C.riparius* embryos. Our data suggest that mesoderm invagination in *C.riparius* increases developmental robustness and efficiency. Additionally, invagination preserves the epithelial character of the mesoderm during internalization and thus decouples epithelial-to-mesenchymal transition in the mesoderm from the positional cell exchange during germband extension. As a result, the anterior-to-posterior order of mesoderm cells may be more effectively preserved throughout gastrulation. During the course of evolution, this could have facilitated the use of positional information of pre-gastrulation blastoderm patterning for post-gastrulation differentiation of the internalized mesoderm.

In the following, we are building on our results in *C.riparius* to assess the evolution of highly coordinated mesoderm invagination as described for *D.melanogaster*. Unspecific injection of *fog* and *t48* mRNA, which at best corresponds to a strong maternal loading, had a specific effect on *C.riparius* mesoderm internalization instead of leading to pleiotropic morphogenetic defects (*Figure 5*). This

observation was unexpected and surprising. In order for *C.riparius* embryos to respond to the early ubiquitous expression of *fog* or *t48* in a spatially restricted manner along the ventral midline, the activity of either gene was apparently realized differently in different regions of the *C.riparius* blastoderm embryo. A strong and exclusively ventral response to *fog* and *t48* activity in *C.riparius* could explain such a local change in the mode of morphogenesis in spite of ubiquitous gene expression. Recent work has shown that ancestral modes of gastrulation can be built upon a gene repertoire that is in part distinct from the network known in *D.melanogaster* (*Pers et al., 2016*; *Stappert et al., 2016*), and the local response to ubiquitous *fog* and *t48* activity in *C.riparius* embryos may thus be elicited by an unknown *fog/t48* sensitive pathway under control of *twist* and/or *snail*. Alternatively, ectopic *fog* and *t48* in *C.riparius* act through a conserved genetic cascade present also in *D.melanogaster*. In this case, the activity of at least one component of this pathway downstream of *fog* and *t48* would have to be locally restricted to the ventral blastoderm. A possible candidate for such a conserved component with spatially restricted activity downstream of *fog* and *t48* is the G-protein coupled receptor Mist. In embryos of both species, *C.riparius* and *D.melanogaster*, *mist* is expressed in a narrow domain along the ventral midline (*Figure 3F,F',N,N'*).

Mist transduces signaling of the extracellular ligand Fog to the intracellular receptor-coupled G-protein $G\alpha_{12/13}$ Cta, which during *D.melanogaster* mesoderm invagination results in apical RhoGEF2 activation. RhoGEF2 is intracellularly enriched at the apical cell membrane by its transmembrane anchor T48, and apical RhoGEF2 activation subsequently results in constriction of the apical actomyosin network and thus coordinated cell shape changes that lead to ventral invagination (*Kerridge et al., 2016*; *Manning et al., 2013*; *Parks and Wieschaus, 1991*; *Dawes-Hoang et al., 2005*; *Morize et al., 1998*; *Barrett et al., 1997*; *Häcker and Perrimon, 1998*; *Fox and Peifer, 2007*; *Kölsch et al., 2007*). In blastoderm embryos of *D.melanogaster*, *fog* and *mist* are expressed in partially overlapping domains, and tissue invagination is observed only where the expression of both genes coincides (*Manning et al., 2013*). Similarly, the activity of ectopic ubiquitous *fog* and *t48* expression in *C.riparius* appears to be limited to a ventral domain that overlaps with endogenous *Cri-mist* expression.

Based on the expression of *Cri-cta* and *Cri-mist* and the induction of mesoderm invagination through ectopic *fog* and *t48*, we speculate that mesoderm ingression as an ancestral mode of cell internalization is based on comparatively low-level *RhoGEF2* activity (*Figure 7A*). This low level of RhoGEF2 activity may be invoked by scattered expression of the ligand Fog, or, as has been suggested before, by low-frequency self-activation of Mist (*Manning and Rogers, 2014*). Our results in wildtype *C.riparius* embryos suggest that under such conditions, cells undergo an early epithelial-to-mesenchymal transition and leave the blastoderm stochastically due to individual rather than collective cell shape changes. This cell behavior changes under conditions that increase RhoGEF2 activity at the apex of mesoderm cells, which may be achieved in two ways: either non-cell-autonomously, by increasing GPCR signaling strength through elevated levels of the secreted ligand Fog, or cell-

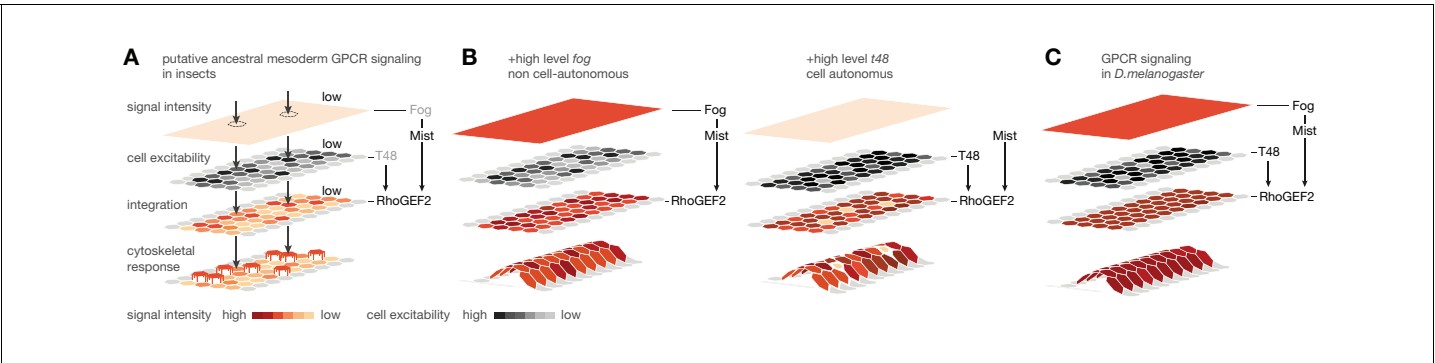

**Figure 7.** Evolutionary scenario for cell-biological changes underlying the origin of coordinated mesoderm invagination in *D.melanogaster*. (A) Cartoon model of putative ancestral mesoderm GPCR signaling based on findings in *C.riparius*. According to the model, ancestral mesoderm ingression in insects is characterized by low-level RhoGEF2 activity in the absence of high level *fog* and *t48* activity. (B,C) Evolutionary gain of either high *fog* or high *t48* expression levels elevates RhoGEF2 activity (B) and thus invokes tissue-level coordination of mesoderm cell behavior as in *D.melanogaster* (C).

autonomously, by improving GPCR signaling efficiency through the apically enriched membrane anchor T48 (*Kölsch et al., 2007*). It is tempting to speculate that nature used similar changes during fly evolution (*Figure 7B*), which ultimately led to the highly efficient GPCR signaling in the ventral blastoderm of *D.melanogaster* (*Figure 7C*).

Although *C.riparius* and *D.melanogaster* diverged from a last common ancestor about 250 million years ago (*Wiegmann et al., 2011*), our functional analysis of mesoderm internalization in both species suggests that surprisingly simple genetic changes are sufficient to reversibly switch between cell ingression and tissue invagination. Without an overt genetic directionality, mesoderm formation in flies has thus evolved either from a Drosophila-like ancestor that essentially lost maternal as well as zygotic *fog* and *t48* expression in *C.riparius*, or from a Chironomus-like ancestor that gained early embryonic *fog* and *t48* expression in successive steps leading to the developmental network described for *D.melanogaster*. We cannot exclude that *C.riparius* evolved from a Drosophila-like ancestor by a secondary loss of early *fog* and *t48* expression, and recent lineage-specific gene duplications, resulting, e.g., in two *fog* copies or *pangolin* (*Klomp et al., 2015*), suggest that *C.riparius* may have diverged significantly from the last common ancestor of flies. Overall however, we favor the hypothesis that *C.riparius* rather than *D.melanogaster* is more reminiscent of ancestral mesoderm internalization in flies. Supporting this view, mesoderm ingression has been reported for an independent family at the base of the insect order of flies (Diptera), *A.gambiae* (*Goltsev et al., 2007*), in the beetle *Tribolium castaneum* (*Handel et al., 2005*), and reflects the general view that mesoderm formation in insects evolved from an ancestor with an ingression-like mode of cell internalization (*Roth, 2004*).

Experimental induction of mesoderm invagination by early *fog* or *t48* expression in *C.riparius* embryos suggests that the evolutionary origin of the *D.melanogaster* gene network is linked to the gain of a ubiquitous maternal and a ventral zygotic enhancer for *fog* and *t48*. Our results in *C.riparius* suggest that high levels of maternal-like ubiquitous expression of *fog* and *t48* are sufficient to invoke mesoderm invagination. Consistent with a critical role of maternal *fog* and *t48* in the evolutionary origin of mesoderm invagination, both genes are also expressed ubiquitously in pre-blastoderm embryos of *D.melanogaster* (*Costa et al., 1994*; *Strutt and White, 1994*) (*Figure 3—figure supplement 2*). However, so far only a weak maternal contribution has been functionally demonstrated for *fog* (*Zusman and Wieschaus, 1985*), and the main function of both genes in *D.melanogaster* mesoderm invagination is derived from their zygotic, twist-dependent ventral expression (*Costa et al., 1994*; *Kölsch et al., 2007*). Thus, mesoderm invagination in *D.melanogaster* originated by the origin of highly specific Twist-dependent enhancers in *fog* or *t48* and later the gain of maternal expression in both genes, or vice versa. We favor the idea that the origin of mesoderm invagination was built on the gain of a maternal enhancer, which may evolve more likely *de novo* from random sequence turnover.

*fog* and *t48* may have acquired their role in coordinating cell behavior during epithelial morphogenesis coincidently with the evolution of fly gastrulation. However, in *D.melanogaster*, the functions of *fog*, *mist*, and RhoGEF2 do not appear to be limited to gastrulation; the genes have been additionally implicated in gastrulation-like folding of epithelia during wing, leg, and salivary gland morphogenesis (*Manning et al., 2013*; *Nikolaidou and Barrett, 2004*; *Ratnaparkhi and Zinn, 2007*), and the presence of *fog* and *t48* orthologues in insect genomes overlaps with the phylogenetic occurrence of winged appendages and salivary glands (*Krenn and Aspöck, 2012*) (*Figure 3—figure supplement 1*). We therefore favor a model in which *fog* and *t48* have played a role in epithelial morphogenesis, at least during late stages of development, long before the origin of mesoderm invagination. Both genes were then subsequently recruited for a function in modern, Drosophila-like gastrulation with a delayed epithelial-to-mesenchymal transition. Such recruitments may occur independently and repeatedly, and it will be informative to study a possible role of *fog* and *t48* in hymenopterans. Species in this insect order also internalize their mesoderm in a coordinated way, but as a stiff, coherent plate along the ventral midline (*Fleig and Sander, 1988*; *Roth, 2004*; *Sauer, 1954*).

Recruiting genes with roles in later developmental stages into the maternal germline may provide the missing link between the divergence seen in late phases of a species' development with diversity that appears earlier – often referred to as a developmental 'hour glass' (*Duboule, 1994*; *Raff, 1996*; *Kalinka and Tomancak, 2012*). This provides a simple, and possibly non-gradual, evolutionary path toward altering early embryonic stages prior to the onset of zygotic transcription, and it

may reflect a more general mechanism that facilitated the diversification of early embryonic development.

# Materials and methods

## Fly stocks

The laboratory culture of *Chironomus riparius* (Meigen) originates from the culture of Gerald K. Bergtrom (University of Wisconsin, Milwaukee, WI) and was obtained from Urs Schmidt-Ott (The University of Chicago, Chicago) (*Klomp et al., 2015*). The culture was maintained as previously described at 25°C and a constant 17/7-hr day/night cycle (*Caroti et al., 2015*). For *Drosophila melanogaster*, the sequenced reference from Kyorin University (y[1],oc[R3.2]; Gr22b[1], Gr22d[1], cn[1], bw[1], sp[1]; LysC[1], lab[R4.2], MstProx[1], GstD5[1], Rh6[1]) was used for wild type analysis and fog; hkb::fog (*Seher et al., 2007*) to analyze fog loss-of-function in the mesoderm.

## Cloning and RNA synthesis

*Cri-cta* (KX009472), *Cri-fog1* (KX009473), *Cri-fog2* (KX009479), *Cri-mist* (KX009474), *Cri-RhoGEF2* (KX009475), *Cri-sna* (KX009476), *Cri-t48* (KX009477) and *Cri-twi* (KX009478) were identified from transcriptome sequences and cloned after PCR amplification from cDNA.

Full length coding sequences of *fog* and *t48* were amplified by PCR from pB26H (*Costa et al., 1994*) and pNB40 (*Kölsch et al., 2007*), CDSs of *Cri-fog1* and *Cri-t48* were amplified by PCR from cDNA. Fragments were cloned into the expression vector pSP35T (*Amaya et al., 1991*) using an adapted *C.riparius* specific Koszac sequence (*Klomp et al., 2015*) to generate capped mRNAs using the mMessage mMachine Kit with SP6 RNA polymerase (Ambion, NY). Synthesized mRNA was dissolved in $H_2O$.

Double-stranded RNAs (dsRNAs) was synthesized essentially as described (*Stauber et al., 2000*) on templates that were amplified by PCR from TOPO-TA-pCRII (Life Technologies, NY) plasmids using vector-specific primers with attached T7 promoter sequences; dsRNAs comprised the following gene fragments (pos. 1 refers to first nucleotide in ORF): *Cri-fog2*, pos. −92 to 1739; *Cri-mist*, pos. +137 to +1111; *Cri-sna*, pos. +201 to +1008; *Cri-twi*, pos. +114 to +1107; *t48*, pos −527 to +1801.

## Injections

Embryos were collected, prepared for injection, and injected essentially as described (*Caroti et al., 2015*). Embryos were injected before the start of cellularization (approximately four hours after egg deposition), and then kept in a moist chamber until the onset of gastrulation. Throughout all procedures, embryos were kept at 25°C (± 1°C). Owing to their small size, *C.riparius* embryos (200 μm length) were always injected into the center of the yolk (50% of anterior-posterior axis). Embryos were heat-fixed using previously established protocols (*Rafiqi et al., 2011*). Embryos were injected with dsRNA typically at concentrations of 300 to 700 ng/μl; mRNA was injected at 0,16 μg/μl (*fog; Cri-fog*) and 1,1 μg/μl (*t48; Cri-t48*), which corresponded to about 0,2 mol/l of *Cri-fog* transcripts and 2 mol/l of *Cri-t48* transcripts.

## qRT-PCR

The efficiency of transcript knockdown by RNAi was evaluated by qRT-PCR as described previously with minor changes (*Stappert et al., 2016*). Total RNA of *C.riparius* knockdown and wildtype embryos at stages ranging from cellular blastoderm to onset of gastrulation was isolated and transcribed using SuperScript^TM III RT (Invitrogen). Transcript quantities were compared between injected and wildtype embryos using SYBR Green (Thermo Scientific). The qPCR reaction was performed according to the manual (SYBR Green PCR master mix, ThermoScientific). The cycling profile was adjusted to 95°C 15 min, 40 cycles 95°C 15 s, 55°C 15 s, 72°C, 15°C and extension at 72°C for 7 min; a melting curve program was added. For the calculation of the RNA ratios the delta delta Ct method was used. Primers used were 5'-CCGGAAAAATATTCCAACGA/5'-GCGAGTGTTGCAATCA-GAAA for *Cri-mist*, 5'-GTGACTTCGAGCTCGCTCTT/5'-CGACGACAACAACAACAACC for *Cri-fog2*, and 5'-AGAAGGCAAAGCTGATGGAA/5'-GGAGCGAAAACAACAACCAT for *Cri-EF1α* as reference.

## Immunohistochemistry

For whole mount *in situ* hybridization, antibody, and nuclear staining, embryos were heat fixed and devitellinized using 1+1 n-heptane and methanol as described (*Rafiqi et al., 2011*). For staining with phalloidin, embryos were fixed using 4% formaldehyde and devitellinized by substituting methanol with 95% ethanol (*Mathew et al., 2011*).

RNA probe synthesis, whole mount *in situ* hybridization, and detection was carried out as described (*Lemke and Schmidt-Ott, 2009*). The following cDNA fragments were used as probes: (pos. 1 refers to first nucleotide in ORF): *cta*, pos +401 to +1180; *Cri-cta*, pos +104 to +982; *Cri-fog1*, pos −9 to +1926; *Cri-fog2*, pos −92 to +1739; *Cri-mist*, pos +137 to +1111; *Cri-sna*, pos +201 to +1008; *Cri-t48*, pos −413 to +1765; *Cri-twi*, pos +114 to +1107; *fog*, pos −38 to +2804; *mist*, pos +518 to +1317; *sna*, pos −146 to +1515; *t48*, pos −527 to +1801; *twi*, pos −376 to +2249.

Antibody and nuclear staining was carried out as described (*Jimenez-Guri et al., 2014*) with modifications: before the staining procedure, embryos were rehydrated after storage in methanol in PBT (0.1% Tween 20 in PBS), digested with Proteinase K (1:2500, 0.08 U/ml; Invitrogen) for 2 min at room temperature, immediately washed with PBT, and fixed in 5% formaldehyde for 25 min. Phosphorylated histone H3 detected using p-Histone H3 antibody (1:100; Santa Cruz Biotechnology) as primary and anti-rabbit Alexa488 (1:200; Jackson Immuno Research) as secondary antibody; nuclei were stained using 4′,6-diamidino-2-phenylindole (DAPI). Phalloidin staining was performed as described (*Panfilio and Roth, 2010*) with modifications (1:200; 200 units/ml stock, Life technologies) in PBT for two hours at room temperature.

## Microscopy

To preserve shape and allow for free rotation along the anterior-to-posterior axis, *D.melanogaster* embryos were mounted in glycerol (in PBS, v/v; *D.melanogaster* 75%, *C.riparius* 50%) with support for the cover glass. Histochemical staining was recorded with DIC on a Zeiss Axio Imager M1 using 20x (dry, 20x/0.8) for *C.riparius* and both 20x and 10x (dry, 10x/0.45) for *D.melanogaster*; fluorescent staining was recorded by single-photon confocal imaging on a Leica system (DMI 4000 B and SP8) using a 40x oil objective (HC PL APO CS2 40x/1.3) for *C.riparius* and a 20x immersol objective (HC PL APO CS2 20x/0.75) for *D.melanogaster* embryos. Image stacks were acquired with a voxel size of 0.28 × 0.28 × 0.42 µm for *C.riparius* and 0.51 × 0.51 × 0.42 µm for *D.melanogaster* by oversampling in z and comprised at least 30–40% of embryo depth orthogonal to the ventral midline. A bright-field lateral view of each analyzed embryo was recorded for developmental staging. Images and stacks were processed using Fiji (2.0.0-rc-34/1.50a), Matlab (R2014b), and Vaa3d (2.801) and assembled into figures in Adobe Photoshop and Adobe Illustrator.

## Image processing

Each embryo was imaged as confocal *z*-stack with three 8-bit channels containing immunofluorescence of pHis H3, DAPI, and/or phalloidin. The image stacks were aligned to have the embryo ventral midline run in parallel to the x axis. From the aligned stacks cell positions, a boundary of the ventral epithelium, and an outline of the egg shell were extracted.

## Reconstruction of cell position, ventral epithelial boundary, and egg shell

### Cell position

To obtain cell positions, cells or cell nuclei were segmented and extracted as objects using *Pixel Classification+Object Classification* of the *Ilastik* framework (*Linux version 1.1.5*) (*Sommer et al., 2011*). The spatial position of each cell within the imaged volume was defined by the center of mass (centroid) of the *iLastik*-segmented cell or its nucleus.

### Ventral epithelium

To obtain a boundary for the ventral epithelium, the staining of individual nuclei in a confocal z stack was blurred into a layer of connected nuclei by applying a Gaussian blur (sigma = 3) and enhancing image contrast (0.35 saturation). This volume was segmented and extracted as a single connected component using manual thresholding. The resulting object was loaded as point cloud into Matlab and remaining holes were closed using *fillholes3d* (maximal gap 20px). For this now closed object, a

surface mesh was generated using *vol2surf*. Both, *fillholes3d* and *vol2surf* functions are part of the matlab iso2mesh toolbox. The surface mesh was separated into an upper mesh (facing the yolk side of the embryo) and a lower mesh (facing the egg shell). The lower mesh, corresponding to the apical side of the nucleus layer, was used approximation of the ventral epithelium and to classify tissue integrity.

### Egg shell

To approximate the egg shell outline in the absence of the vitelline membrane after fixation and devitellinization, a 3D alpha hull mesh was cast over the centroids of the segmented nuclei. Using the Matlab function *alphavol* with the *xyz* coordinates of the centroids (probe radius R set to default), this approach yielded the convex hull, a surface that enclosed all ventral nuclei with the smallest possible area and thus resulted in a virtual envelope that leveled out indentations and ventral furrow invagination. To compensate for the offset between nucleus center and egg shell edge, the alpha hull mesh was uniformly expanded by half a nucleus diameter in the ventral direction.

## Quantification of morphological parameters

The developmental progression of fly gastrulation was assessed by the degree of germ band extension in lateral mid-sections of bright-field images. The deepest indentation of the infolding posterior midgut was taken as reference to the posterior end of the germband, and germband extension was measured in percent egg length relative to the anterior-posterior axis of the embryo (% GBE, *Figure 1F*). To quantify morphological features of mesoderm formation, a region of interest (ROI) was defined by projection of a rectangle that encompassed all nuclei in the ventral 30% of the embryonic circumference and from 25% to 75% egg length along the anterior-to-posterior axis.

Coherence of mesoderm cell behavior along the anterior-posterior axis was qualitatively assessed in individual embryos by nuclear centre-lines. Centre-lines were generated for each transversal section of the ROI by determining the shortest path from the left to the right side of the ventral epithelium and successively passing through the centre of mass of each nucleus. The stack of center-lines was used as a visual and qualitative measure to judge coherence along the anterior-to-posterior of individual embryos. To compare tissue coherence between species, corresponding stages of mesoderm internalization were identified according to germband extension (*Figure 1F*).

Morphological parameters of mesoderm formation were quantified within the ROI by determining the distance of cells to the epithelial surface of the embryo and the egg circumference using Matlab scripts (https://github.com/lemkelab/mesoderm). The ROI was sampled by nine equal-sized intervals, each containing approximately 30 cells. Based on empirical thresholding, a cell was classified as internalized if the distance between its nucleus center and the reconstructed epithelial surfaceegg shell exceeded the length of one cell nucleus; it was independently classified as ingressed if the distance between nucleus center and the reconstructed reconstructed epithelial surface exceeded the length of one cell nucleus. Morphological parameters were then defined as follows: *internalization* - percentage of internalized nuclei in ROI; *maximal cell depth* - maximal measured distance between any nucleus center and the reconstructed egg shell in a transversal section; *integrity*: percentage of internalized cells that are not ingressed; *furrow depth*: maximal measured distance between internalized nucleus center and reconstructed egg shell in a transversal section. The *furrow width* of the respective cross sections was measured manually as a percentage of the embryo width directly on the *zy* image stacks in *Fiji*. As our definition of *integrity* as quantitative measure was measured only for internalized cells, this method cannot directly account for cases where cells stay within an epithelium in close contact with the egg outline (e.g. *Cri-sna* RNAi) In this case, the quantification of integrity was extended to all cells in ROI, regardless of internalization.

## Principal component analysis

The principal component analysis included all transversal sections for *C.riparius* and *D. melanogaster*, each described by its values for the five parameters: *internalization, maximal cell depth, integrity, furrow depth* and *furrow width*. All parameters were normalized by z-score normalization. Based only on parameter values and covariance, the analysis identified the two main axes along which variation in our data set was maximal. For all wild type sections, the resulting two principal components (PC1 and PC2) accounted for 57% and 21% of variance, respectively. A biplot for the

PCA was generated to visualize the contribution of each of the parameters to the two principal components. To test whether parameter separation was dependent on variation in space (i.e. the position of a given transversal section along the anterior-to-posterior axis of an embryo) and developmental age (as measured by % of GBE), we color-coded cross sections according to their position and developmental age.

## Quantification of variation and efficiency

Analysis of within and between-embryo variation was performed on all transversal sections and in the z-score normalized time-independent morphospace for *C.riparius* defined by *internalization*, *epithelial integrity*, *furrow depth* and *maximal cell depth*. Variation in cell behavior between transversal sections was calculated as Euclidean distance between transversal sections in morphospace. Variation between groups (H$_2$O; +*fog* / +*t48*) was measured as the average distance between all cross sections of all embryos within this group. Variation between embryos was measured as the average distance of the cross sections of one embryo to the cross sections of all other embryos in one group. Variation within an embryo was measured as the standard deviation from the average distance of all cross sections within a given embryo. Efficiency of mesoderm internalization was measured by the number of internalized cells in embryos with 15 to 25% GBE.

## Statistics

Statistical comparisons of distributions were performed as comparison of their medians via the *Matlab* implementation of the Wilcoxon rank sum test, as most variables analyzed were not normally distributed. p-values of this test are indicated in the figure legend.

## Phylogenetic occurrence of *D.melanogaster* mesoderm genes in insects

Genes were considered as present in the last common ancestor of insects if they were included in the insect as well as the metazoan or arthropod OrthoDB orthology group (*Waterhouse et al., 2013*). In addition, *t48* and *fog* sequences were identified by recursive searches of insect genomes with a position-specific scoring matrix (pssm). The pssm was generated in a psi-blast (*Altschul et al., 1997*) against non-redundant translated genome databases of representative insect genomes using two iterations and standard parameters. To identify orthologue candidates, this pssm was used in a psi-blast of individual insect genomes with an e-value cut-off at 0.001. For each searched genome, the top three contigs were selected and submitted to a reciprocal blast to identify the closest homologous gene match as well as putative species-specific gene duplications. In the first iteration, the pssm was generated using the *D.melanogaster* proteins T48 (GenBank, CAA55003.1) and Fog (GenBank, NP_523438), and the reciprocal blast was performed in the translated *D.melanogaster* transcriptome (BDGP5). In each following iteration, the pssm was generated using the closest homologous gene match identified in the species that was most closely related to the species used to generate the pssm in the previous search. Major taxa were represented by available genome (g) and transcriptome (t) sequence assemblies: Chelicerata (*Limulus polyphemus*, g), Myriapoda (*Strigamia maritima*, g) Crustaceans (*Daphnia pulex*, g), Diplura (*Catajapyx aquilonaris*, g), Odonata (*Ladona fulva*, g), Ephemeroptera (*Ephemera danica*, g), Orthoptera (*Locusta migratoria*, g), Phasmatodea (*Timema cristinae*, g), Blattodea (*Blattella germanica*, g), Isoptera (*Zootermopsis nevadensis*, g), Thysanoptera (*Frankliniella occidentalis*, g), Hemiptera (*Oncopeltus fasciatus*, g; *Acyrthosiphon pisum*, g), Psocodea (*Pediculus humanus*, g), Hymenoptera (*Apis mellifera*, g; *Nasonia vitripennis*, g), Megaloptera (*Sialis lutaria*, t), Neuroptera (*Chrysopa pallens*, g), Strepsiptera (*Mengenilla moldrzyki*, g), Coleoptera (*Tribolium castaneum*, g), Trichoptera (*Limnephilus lunatus*, g), Lepidoptera (*Bombyx mori*, g; *Papilio polytes*, g), Siphonaptera (*Archaeopsylla erinacei*, t), Diptera (*Chironomus riparius*, g; *Megaselia abdita*, g; *Drosophila melanogaster*, g). Assemblies were obtained from insect base (*Yin et al., 2016*), except for *L.polyphemus* (GenBank, GCA_000517525.1), *S.maritima* (GenBank, GCA_000239455.1), *D.pulex* (GenBank, GCA_000187875.1), *C.riparius* (Genbank, GCA_001014505.1), *M.abdita* (GenBank, GCA_001015175.1), and *D.melanogaster* (flybase, 6.07). Non-redundant protein BLAST databases were generated for each of the genome assemblies by translating the DNA sequence into amino acid sequences using *transeq* (EMBOSS; *Rice et al., 2000*) and the NCBI BLAST+ suite (*Camacho et al., 2009*).

## Acknowledgements

We thank U Schmidt-Ott for the Chironomus culture; M Leptin for plasmids and Drosophila stocks; T Sandmann and M Bernardo for technical help and discussions; L Centanin, A Guse, T Holstein, U Schmidt-Ott, J Wittbrodt, and members of the Lemke lab for discussions and comments on the manuscript; J Großhans, C Lye, and S Roth for helpful reviews; R Hodge for manuscript editing; I Lohmann, J Lohmann and J Wittbrodt for sharing laboratory equipment; and J Wittbrodt for continuous and generous support. Funded by DFG grant LE 2787/1-1.

## Additional information

### Funding

| Funder | Grant reference number | Author |
|---|---|---|
| Deutsche Forschungsgemeinschaft | LE 2787/1-1 | Steffen Lemke<br>Silvia Urbansky<br>Paula González Avalos<br>Maike Wosch |

The funders had no role in study design, data collection and interpretation, or the decision to submit the work for publication.

### Author contributions

SU, Conception and design, Acquisition of data, Analysis and interpretation of data, Drafting or revising the article; PGA, SL, Conception and design, Analysis and interpretation of data, Drafting or revising the article; MW, Acquisition of data, Contributed unpublished essential data or reagents

### Author ORCIDs

Steffen Lemke, http://orcid.org/0000-0001-5807-2865

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
