## [Decision Letter]

Thank you for submitting your article "A genetic switch controls the mode of gastrulation" for consideration by *eLife*. Your article has been reviewed by three peer reviewers, one of whom, Siegfried Roth, is a member of our Board of Reviewing Editors and the evaluation has been overseen by Detlef Weigel as the Senior Editor. The following individuals involved in review of your submission have agreed to reveal their identity: Jörg Großhans (Reviewer #2); Claire Lye (Reviewer #3).

The reviewers have discussed the reviews with one another and the Reviewing Editor has drafted this decision to help you prepare a revised submission.

The paper deals with the molecular mechanisms contributing to the evolution of mesoderm internalization in insects. The authors compare two dipteran species: *Drosophila melanogaster* and Chironomus riparius. The system provides a rare opportunity to address a macro-evolutionary question with an experimental approach. All reviewers agreed that this is an interesting study, which is important for the field of developmental biology as it exemplifies how the rich knowledge of pathways and genes controlling morphogenesis in *Drosophila* can be employed to explain the variations in visible morphology between closely and more distantly related species in terms of defined genetic variations.

However, there are a number of issues that need to be addressed prior to publication. Most of these issues concern the presentation of the study. There is only one point (9) in which an additional experiment was suggested.

1) The title should be more specific. The topic is mesoderm internalization and not gastrulation. The term genetic switch is misleading and in the context of the paper not appropriate.

2) Several terms are used for the process and types of mesoderm internalization, e.g. stochastic translocation, stochastic cell migration, stochastic cell internalization, stochastic migration of individual cells, cell ingression, single cell migration, stochastic internalization. We suggest authors use internalization as a general term, ingression for Chironomous, and invagination for *Drosophila*.

3) Mesoderm invagination is based on cell shape changes and cell movement. The authors analyzed fixed embryos stained for F-actin and DNA. In Figure 2—figure supplement 1, the authors present data that are supposed to show that nuclear position is a good proxy for cell position. This is convincing in the example they present, but we would like to see examples for both C.riparius and D.*melanogaster*, and at a time when mesoderm internalization is more progressed. It is important to demonstrate that this way of measuring cell position holds for the full range of samples analyzed.

4) The Principal Component Analysis does not seem to be mentioned in the methods and generally needs more explanation. Do any statistical methods underlie the assertions that the 2 species separate most sensitively along epithelial integrity, or is this based on interpretation of the graph? Some details of this in the main text would be helpful.

5) In Figure 2 it would be nice to see statistics showing where differences are significant.

6) In all RNAi experiments (Figure 4) it would be nice to see statistics showing where differences are significant.

7) Results section – twi RNAi: "[…]internalization of cells outside the ventral midline[…]". This is not like *Drosophila* (as the authors claim) and suggests some non-autonomous effects, which spread from the mesodermal region to neuroectoderm and are suppressed in the presence of Cr-twi.

8) Results section – maternal loading: While for Dm-fog a weak maternal contribution has been functionally demonstrated (Zusman and Wieschaus, 1985), we are not aware of a maternal contribution for t48. However, even if this were the case for t48, the main function for both fog and t48 is derived from the zygotic ventrally localized expression in *Drosophila*. The genetics are very clear!

9) Results section – Expression of Cr-fog2 and Cr-mist: It is rather astonishing that a fog paralog and the receptor mist are expressed ventrally prior to gastrulation. Indeed KD of fog2 seems to have a weak effect on mesoderm internalization (Figure 4 compared to Figure 4). Are the authors sure the RNAi knockdown of these genes is working? Ideally, the authors should perform RT-PCR demonstrating a significant knockdown, which is important for the assertion the fog signalling pathway does not play a role in gastrulation in wildtype C.riparius

10) In the last part of the study, the authors would like to address the question whether epithelial infolding would make mesoderm invagination more robust than by cell ingression. For this they apply wounds by microinjection and assay developmental variation. This is done in normal and "rescued" Chironomus embryos. We do not find the assay robust and the data convincing as the read-out of the assay is not clearly defined. In our view this experiment could be removed entirely without affecting the strength of the study.

11) Discussion and conclusion section – Gain of a maternal enhancer for fog and t48: Since it is very clear that the contribution of fog and t48 to gastrulation in *Drosophila* depends on their ventral zygotic expression we do not believe that the gain of maternal enhancers was the crucial evolutionary change. It is rather surprising that the Chironomus injections of t48 and fog mRNA showed such specific effects and did not result in pleiotropic morphogenetic defects. Regarding the injection experiments we think you should use a term like 'ubiquitous ecotopic expression' rather than mimicking maternal loading.

12) In reference to the Discussion, I am not convinced there is sufficient evidence to claim that RhoGEF2 activation normally occurs at low levels in C.riparius, though expression patterns of cta and mist might be suggestive if this. For such a claim to be tested further knockdown experiments would be required. The alternative that in wildtype C.riparius mesoderm internalisation could be controlled by a separate pathway, but that it is competent to response to ectopic fog and T48 should also be discussed.

13) The authors propose that the ingression mode of Chironomus is the evolutionary older mode and that the epithelial infolding mode has emerged later in evolution by gain of T48 and Fog expression in the mesoderm anlage. This model is not fully consistent with the situation on the genetic level. T48 and Fog genes are present in both species. No arguments are presented that would address the directionality, whether *Drosophila* has gained the expression or whether Chironomus has lost the expression of T48/Fog in the mesoderm anlage. The argumentation that Chironomus has lost the expression of T48 and Fog in the mesoderm anlage is equally valid. The authors should at least discuss the alternative model that the ingression mode in Chironomus is derived from an epithelial infolding by loss of Fog and T48 expression.

14) The manuscript contains nine supplemental figures. The authors should incorporate most if not all of the supplemental data into the main body of the manuscript. For example, the manuscript should start with a comparative description of mesoderm invagination in *Drosophila* and Chironomus that also defines the staging that is used furtheron in the study. The current Figure 1 is not sufficient for such a presentation, as the staging is not clear and no cells are shown. For example, in Figure 1/Chironomus, it is not clear whether only nuclei shifted their position or whether cell ingression has started. Multiple stages together with a time scale should be presented to describe the dynamics of the process. Much of this is shown in Figure Figure 1—figure supplement 1.

15) This paper contains analysis of a large quantity of quantitative data. Supply of underlying data would be useful for verification of their findings and may be of use to other researchers undertaking related studies.

---

## [Author Response]

*The paper deals with the molecular mechanisms contributing to the evolution of mesoderm internalization in insects. The authors compare two dipteran species: Drosophila melanogaster and Chironomus riparius. The system provides a rare opportunity to address a macro-evolutionary question with an experimental approach. All reviewers agreed that this is an interesting study, which is important for the field of developmental biology as it exemplifies how the rich knowledge of pathways and genes controlling morphogenesis in Drosophila can be employed to explain the variations in visible morphology between closely and more distantly related species in terms of defined genetic variations.*

However, there are a number of issues that need to be addressed prior to publication. Most of these issues concern the presentation of the study. There is only one point (9) in which an additional experiment was suggested.

We appreciate the helpful and supportive comments provided by the reviewers, and we feel they have improved the manuscript substantially. Specific changes in the manuscript have been indicated wherever possible, with the Discussion being an exception: based on the comments, we felt that a restructured Discussion would better be able to address the issues raised and, overall, improve readability. Likewise, the first two figures have been substantially reorganized and now include most of the data that has been previously placed in the supplements.

*1) The title should be more specific. The topic is mesoderm internalization and not gastrulation. The term genetic switch is misleading and in the context of the paper not appropriate.*

We have changed the title to be more specific. It now reads: “Fog and t48 drive the evolution of coordinated mesoderm internalization in flies”.

*2) Several terms are used for the process and types of mesoderm internalization, e. g. stochastic translocation, stochastic cell migration, stochastic cell internalization, stochastic migration of individual cells, cell ingression, single cell migration, stochastic internalization. We suggest authors use internalization as a general term, ingression for Chironomous, and invagination for Drosophila.*

We welcome the suggestion and adopted a more coherent terminology. The terms internalization, ingression, and invagination are now introduced in the context of fly mesoderm formation: “… we have studied the evolutionary transition between two distinct modes of cell internalization during fly gastrulation, i.e. mesoderm invagination in the fruit fly *Drosophila melanogaster* and mesoderm ingression in the midge *Chironomus riparius* (Figure 1)”. Either term is defined by example in Figure 1 and then used to describe the morphogenetic processes in *C.riparius* and *D.melanogaster* throughout the remaining text.

*3) Mesoderm invagination is based on cell shape changes and cell movement. The authors analyzed fixed embryos stained for F-actin and DNA. In Figure 2—figure supplement 1, the authors present data that are supposed to show that nuclear position is a good proxy for cell position. This is convincing in the example they present, but we would like to see examples for both C.riparius and D.melanogaster, and at a time when mesoderm internalization is more progressed. It is important to demonstrate that this way of measuring cell position holds for the full range of samples analyzed.*

Following this suggestion, we have added additional data and changed Figure 1 as follows: panel A and B now contain a representative transversal section each of C.riparius and D.*melanogaster* embryos during the progression of mesoderm internalization; panel C depicts nuclei and cell outlines of the internalizing mesoderm cells based on DNA and F-actin stain. For both stainings, we have determined the cell center (centroids) and calculated the deviation between the two methods. Panel D shows that, for the analysis of whole embryos, the deviation of the two methods is less than 10% of cell height for almost all internalizing cells (97% in C.riparius; 90% in D.*melanogaster*).

*4) The Principal Component Analysis does not seem to be mentioned in the methods and generally needs more explanation. Do any statistical methods underlie the assertions that the 2 species separate most sensitively along epithelial integrity, or is this based on interpretation of the graph? Some details of this in the main text would be helpful.*

A more detailed description of the principal component analysis has been added to the methods. Following main point #14, the analysis is now fully integrated into Figure 2, and the interpretation of the graphs has been integrated in the main text. The relevant section now reads (p3, final paragaph): “To address the extent to which each individual parameter contributed to the global differences in *C.riparius* and *D.melanogaster* mesoderm formation and detect biases in the analysis, we performed a principal component analysis of all the sections and parameters. We found three principal parameters that, regardless of intra-species variation in space and time, permitted the best discrimination between ingression and invagination: cell internalization, the depth of the ventral furrow and epithelial integrity (Figure 2). Accordingly, invagination in *D.melanogaster* was characterized by high tissue integrity, a deep ventral furrow, and a high number of internalized cells, while ingression in *C.riparius* was characterized by low tissue integrity, a shallow ventral furrow, and a lower number of internalized cells. The number of internalized cells as well as the depth of the ventral furrow may be in part depended on the size of the mesoderm anlage and thus contain a species-specific component, whereas integrity was independent of the size of the mesoderm anlage.”

*5) In Figure 2 it would be nice to see statistics showing where differences are significant.*

Statistics were added throughout all analyses to indicate where differences are significant.

*6) In all RNAi experiments (Figure 4) it would be nice to see statistics showing where differences are significant.*

See also response to comment 5: statistics were added throughout all analyses to indicate where differences are significant.

*7) Results section – twi RNAi: "[…]internalization of cells outside the ventral midline[…]". This is not like Drosophila (as the authors claim) and suggests some non-autonomous effects, which spread from the mesodermal region to neuroectoderm and are suppressed in the presence of Cr-twi.*

We thank the reviewers for pointing out this passage in which we were using “ventral midline” in a poorly defined manner. We have revised the sentence, which now reads (p.4): “The knockdown of *Cri-twi* by RNAi resulted in embryos that lacked a visible ventral groove. Despite the *Cri-twi knockdown*, nuclei of internalized cells could still be observed within the ventral-most 10-15% of the embryonic circumference (Figure 4), which corresponded to a slightly narrower domain than that of wildtype *Cri-twist* expression.”

*8) Results section – maternal loading: While for Dm-fog a weak maternal contribution has been functionally demonstrated (Zusman and Wieschaus, 1985), we are not aware of a maternal contribution for t48. However, even if this were the case for t48, the main function for both fog and t48 is derived from the zygotic ventrally localized expression in Drosophila. The genetics are very clear!*

The suggested citation for the functional maternal contribution of fog (Zusman and Wieschaus, 1985) has been added in the Results section and the Discussion section. In the Discussion, we emphasize the weak functional contribution of the maternal fog transcript as follows:”… so far only a weak maternal contribution has been functionally demonstrated for *fog* (Zusman and Wieschaus, 1985), and the main function of both genes [fog and t48] in *D.melanogaster* mesoderm invagination is derived from their zygotic, twist-dependent ventral expression (Costa et al., 1994; Kölsch et al., 2007).”

Early ubiquitous expression of t48 prior to blastoderm stages has been first reported by Strutt, D. I. and White, R. A. (1994). Mech Dev 46, 27–39. The citation is included at the same place as the citation of Zusman and Wieschaus (1985). In addition, we have provided modENCODE RNAseq data from flybase and stainings of early ubiquitous expression of *t48* and *fog* by whole mount in situ hybridization in the figure supplement of Figure 3 (Figure 3—figure supplement 1). All data support a maternal contribution of *t48* in *D.melanogaster*.

*9) Results section – Expression of Cr-fog2 and Cr-mist: It is rather astonishing that a fog paralog and the receptor mist are expressed ventrally prior to gastrulation. Indeed KD of fog2 seems to have a weak effect on mesoderm internalization (Figure 4 compared to Figure 4). Are the authors sure the RNAi knockdown of these genes is working? Ideally, the authors should perform RT-PCR demonstrating a significant knockdown, which is important for the assertion the fog signalling pathway does not play a role in gastrulation in wildtype C.riparius*

We thank the reviewers for emphasizing the need to coherently outline our line of thoughts. It was not our intention to make a strong statement that knockdown of *Cri-fog2* or *Cri-mist* does not play any role in wildtype C.riparius gastrulation. We have and still do speculate in the Discussion that they may play a role: “… that mesoderm ingression as a mode of cell internalization is based on comparatively low-level RhoGEF2 activity. This low level of RhoGEF2 activity may be invoked by scattered expression of the ligand Fog, or, as has been suggested before, by low-frequency self-activation of Mist.”

However, the role of *Cri-mist* and *Cri-fog2* appears to be less prominent than in D.*melanogaster*. This interpretation is supported by the very weak effects we are seeing, e.g. after fog2 knockdown experiments, for which we added controls by qPCR (see methods and legend to Figure 4). The corresponding section now reads: “[…] visual inspection of embryos after RNAi knockdown of either gene, […] provided evidence for just a weak phenotype (Figure 4). Following the quantitative analysis, we found that the median cell internalization and furrow depth differed in *Cri-mist* RNAi embryos from water injected embryos. In *Cri-fog2* RNAi embryos, the difference was smaller and significant only for an increase in furrow and maximal cell depth (Figure 4).”

*10) In the last part of the study, the authors would like to address the question whether epithelial infolding would make mesoderm invagination more robust than by cell ingression. For this they apply wounds by microinjection and assay developmental variation. This is done in normal and "rescued" Chironomus embryos. We do not find the assay robust and the data convincing as the read-out of the assay is not clearly defined. In our view this experiment could be removed entirely without affecting the strength of the study.*

We agree with the reviewers that the text could be improved by additional information that defines the read-out of the assay and thus allows to evaluate its robustness. We have provided this information by extending Figure 6, which in its panel A now describes for two selected embryos how variation in mesoderm formation has been quantified. The technical details are outlined in the corresponding section in the Material and methods section. In our view, the assay is robust and provides informative results for the study because it points to potential advantages which the innovation of mesoderm invagination could have had during the course of fly evolution.

*11) Discussion and conclusion section – Gain of a maternal enhancer for fog and t48: Since it is very clear that the contribution of fog and t48 to gastrulation in Drosophila depends on their ventral zygotic expression we do not believe that the gain of maternal enhancers was the crucial evolutionary change. It is rather surprising that the Chironomus injections of t48 and fog mRNA showed such specific effects and did not result in pleiotropic morphogenetic defects. Regarding the injection experiments we think you should use a term like 'ubiquitous ecotopic expression' rather than mimicking maternal loading.*

We followed the suggestion of the reviewers and used ‘ubiquitous expression’ instead of ‘mimicking maternal loading’. The result that Chironomus injections of t48 and fog mRNA showed specific effects and did not result in pleiotropic morphogenetic defects is now discussed in detail in the second and third paragraph of the Discussion. The genetic changes that would be required for the putative transition from an ancestral, Chironomus-like mode of mesoderm ingression to the *Drosophila* mode of mesoderm invagination are also discussed in detail in the Discussion.

*12) In reference to the Discussion, I am not convinced there is sufficient evidence to claim that RhoGEF2 activation normally occurs at low levels in C.riparius, though expression patterns of cta and mist might be suggestive if this. For such a claim to be tested further knockdown experiments would be required. The alternative that in wildtype C.riparius mesoderm internalisation could be controlled by a separate pathway, but that it is competent to response to ectopic fog and T48 should also be discussed.*

Following the suggestion of the reviewers, we have specified our thoughts on low-level RhoGEF2 activity in C.riparius as speculation: “Based on the expression of Cri-cta and Cri-mist and the induction of mesoderm invagination through ectopic fog and t48, we speculate that […]”. The alternative that in wildtype C.riparius mesoderm internalisation could be controlled by a separate pathway is now discussed: “[…] Recent work has shown that ancestral modes of gastrulation can be built upon a gene repertoire that is in part distinct from the network known in D.*melanogaster*, and the local response to ubiquitous fog and t48 activity in C.riparius embryos may thus be elicited by an unknown fog/t48 sensitive pathway under control of twist and/or snail. […]”

13) The authors propose that the ingression mode of Chironomus is the evolutionary older mode and that the epithelial infolding mode has emerged later in evolution by gain of T48 and Fog expression in the mesoderm anlage. This model is not fully consistent with the situation on the genetic level. T48 and Fog genes are present in both species. No arguments are presented that would address the directionality, whether Drosophila has gained the expression or whether Chironomus has lost the expression of T48/Fog in the mesoderm anlage. The argumentation that Chironomus has lost the expression of T48 and Fog in the mesoderm anlage is equally valid. The authors should at least discuss the alternative model that the ingression mode in Chironomus is derived from an epithelial infolding by loss of Fog and T48 expression.

We welcome the suggestion of the reviewers and discuss alternative directions of genetic changes in the Discussion section.

*14) The manuscript contains nine supplemental figures. The authors should incorporate most if not all of the supplemental data into the main body of the manuscript. For example, the manuscript should start with a comparative description of mesoderm invagination in Drosophila and Chironomus that also defines the staging that is used furtheron in the study. The current Figure 1 is not sufficient for such a presentation, as the staging is not clear and no cells are shown. For example, in Figure 1/Chironomus, it is not clear whether only nuclei shifted their position or whether cell ingression has started. Multiple stages together with a time scale should be presented to describe the dynamics of the process. Much of this is shown in Figure 1—figure supplement 1.*

We followed the suggestion by the reviewers and have incorporated most supplemental figures in the main text. Figure 1 has been specifically modified to contain a comparative description of mesoderm internalization in Drosophila and Chironomus, the introduction of nuclei as reliable measure for cell position, as well as the staging scheme to orient the reader with respect to the relative timing of events.

*15) This paper contains analysis of a large quantity of quantitative data. Supply of underlying data would be useful for verification of their findings and may be of use to other researchers undertaking related studies.*

We have provided the quantitative data underlying our analysis for each treatment and embryo in the appendix. For each category (Cri wt, Dme wt, Cri+H20, etc) we provide one text file containing the x,y,z coordinates, distance to cellular surface (d_cell) and distance to ventral surface (d_ventral) for all nuclei within the analyzed region of interest.